# D121 Located within the DRY Motif of P2Y12 Is Essential for P2Y12-Mediated Platelet Function

**DOI:** 10.3390/ijms231911519

**Published:** 2022-09-29

**Authors:** Carol Dangelmaier, Benjamin Mauri, Akruti Patel, Satya P. Kunapuli, John C Kostyak

**Affiliations:** 1Sol Sherry Thrombosis Research Center, Lewis Katz School of Medicine, Temple University, Philadelphia, PA 19140, USA; 2Cardeza Center for Hemostasis, Thrombosis, and Vascular Biology, Department of Medicine, Thomas Jefferson University, Philadelphia, PA 19107, USA

**Keywords:** blood platelets, thrombosis, hemostasis, adenosine diphosphate, platelet aggregation

## Abstract

Platelets are anucleate cells that mediate hemostasis. This occurs via a primary signal that is reinforced by secreted products such as ADP that bind purinergic receptors (P2Y1 and P2Y12) on the platelet surface. We recently identified a human subject, whom we termed platelet defect subject 25 (PDS25) with a platelet functional disorder associated with the P2Y12 receptor. PDS25 has normal blood cell counts and no history of bleeding diathesis. However, platelets from PDS25 have virtually no response to 2-MeSADP (a stable analogue of ADP). Genetic analysis of P2Y12 from PDS25 revealed a heterozygous mutation of D121N within the DRY motif. Rap1b activity was reduced in platelets from PDS25, while VASP phosphorylation was enhanced, suggesting that signaling from the P2Y12 receptor was interrupted by the heterozygous mutation. To explore this further, we produced knock-in mice that mimic our subject. Bleeding failed to cease in homozygous KI mice during tail bleeding assays, while tail bleeding times did not differ between WT and heterozygous KI mice. Furthermore, occlusions failed to form in most homozygous KI mice following carotid artery injury via FeCl_3_. These data indicate that the aspartic acid residue found in the DRY motif of P2Y12 is essential for P2Y12 function.

## 1. Introduction

The primary physiological function of platelets is to act as an essential mediator in maintaining homeostasis of the circulatory system by forming hemostatic thrombi that prevent blood loss and maintain vascular integrity [1,2]. When stimulated with agonists, platelets change their shape, aggregate, and release their granular contents. Release of platelet granular contents includes ADP, which binds to purinergic receptors (P2) and is responsible for potentiation of the initial signal. 

Two families of P2 receptors have been described. The P2X receptors are ligand-gated ion channels, while the P2Y receptors are G protein-coupled receptors (GPCRs) [3]. There are 8 known P2Y receptors expressed in humans, but only the P2Y1 (G_q_) and P2Y12 (G_i_) receptors are found on platelets [3,4]. ADP binding to P2Y1 is essential for platelet shape change, while co-stimulation of P2Y1 and P2Y12 is necessary for ADP-induced platelet aggregation and thromboxane generation [5,6]. We have previously reported that the P2Y12 receptor also potentiates other agonist-induced platelet functional responses and is necessary for Akt activation via other agonists [7,8].

P2Y12 is coupled to G_i_, which inhibits adenylyl cyclase and results in activation of phosphoinositide 3-kinase (PI3K) [9]. The activation of PI3K inhibits the RAP-GAP RASA-3 and allows the formation of GTP-bound Rap1b [10]. Activation of Rap1b is important for integrin activation inside-out signaling, which is essential for hemostasis [11,12,13]. Not surprisingly, some mutations within the P2Y12 genome have resulted in bleeding.

There are several documented examples of human P2Y12 variants which result in different phenotypes from reduced receptor function to a deletion of P2Y12 protein expression. In several cases the groups responsible for detailing these mutations were able to identify new amino acid residues that may be important for P2Y12 function. One patient, who presented with abnormal bleeding, was predicted to have a heterozygous K174E P2Y12 mutation [14]. Platelets from this individual were less sensitive to ADP and it was revealed that the residue in question is important for receptor interaction with ADP. Another documented P2Y12 missense mutation predicted a P341A substitution, which lies in the PDZ binding domain. In this case, P2Y12 protein was expressed normally, but resensitization of the receptor was impaired due to a defect in receptor internalization [15]. Lecchi et al., reported on a homozygous mutation found in 2 brothers, which causes impaired receptor activation [16]. They revealed that H187 is vital for receptor activation as the H187Q mutation found in the 2 brothers permitted ADP to bind P2Y12, but not activate the receptor. Germane to our study is a report concerning a patient with chronic bleeding that had a homozygous R122C substitution within the DRY motif of P2Y12 [17]. There are also several examples of heterozygous and homozygous mutations that result in either reduced or absent expression of P2Y12 [18,19,20].

Here, we report on a subject with no bleeding diathesis who is heterozygous for a D121N substitution within the DRY motif of P2Y12. Platelets isolated from this subject are refractory to 2-MeSADP stimulation even at high doses but respond to stimulation through the G_z_-coupled epinephrine receptor. An analysis of the coding sequence of P2Y12 revealed a G to A SNP that predicts a D to N substitution within the P2Y12 receptor.

## 2. Materials and Methods

### 2.1. Antibodies and Reagents

All reagents were purchased from Thermo Fisher Scientific unless otherwise stated. 2-MeSADP, ADP, and Apyrase (type V) were purchased from Sigma (St. Louis, MO, USA). AR-C69931MX was a gift from the Medicines Company (Parsippany, NJ, USA). AYPGKF was purchased from GenScript (Piscataway, NJ, USA). Phosphorylated Akt S473 and phosphorylated VASP S157 were purchased from Cell Signaling Technologies (Beverly, MA, USA), while the total Akt antibody was purchased from Santa Cruz Biotechnology (Santa Cruz, CA, USA). The total VASP antibody was purchased from Origene (Rockville, MD, USA). Odyssey blocking buffer and secondary antibodies IRDye 800CW goat anti-rabbit and IRDye 680LT goat anti-mouse were purchased from Li-Cor (Lincoln, NE, USA). Collagen and Chronolume, used for the detection of secreted ATP, were purchased from Chrono-Log Corporation (Havertown, PA, USA). GTP-bound Rap1b was assessed using a kit from Cell Signaling and cAMP production was measured using a kit from Enzo. Prostaglandin E1 was also purchased from Enzo (New York, NY, USA). AseI was purchased from New England BioLabs (Ipswitch, MA, USA).

### 2.2. Preparation of Human Platelets

Blood was collected from informed, healthy donors into one-sixth volume ACD (85 mM sodium citrate, 71.4 mM citric acid, and 111 mM dextrose). Human platelets were prepared as previously described [21]. Platelets were resuspended in Tyrodes buffer (137 mM sodium chloride, 2.7 mM potassium chloride, 2 mM magnesium chloride, 0.42 mM sodium phosphate monobasic, 10 mM HEPES, 0.1 U/mL Apyrase, and 0.1% dextrose adjusted to pH 7.4) at a concentration of 2 × 10^8^/mL. Blood counts were collected using a Hemavet blood cell analyzer from Drew Scientific (Miami Lakes, FL, USA).

### 2.3. Platelet Aggregation and ATP Secretion

All platelet aggregation and secretion experiments were carried out using a lumi-aggregometer (Chrono-log) at 37 °C under stirring conditions. Platelet aggregation was measured using light transmission and ATP secretion was measured using Chrono-lume (a luciferin/luciferase reagent).

### 2.4. Western Blotting

Western blotting procedures were performed as described previously [22]. Briefly, platelets were stimulated for the indicated time points in a lumi-aggregometer with either a GPVI or a PAR-4 agonist. The reaction was stopped by precipitating the platelet proteins using 0.6 N HClO_4_ and washed with water prior to the addition of sample loading buffer. Platelet protein samples were then boiled for 5 min prior to resolution by SDS-PAGE and transfer to nitrocellulose membranes. The membranes were then blocked using Odyssey blocking buffer and incubated overnight with primary antibodies against the indicated protein. The membranes were then washed with Tris-buffered saline containing 0.1% tween-20 prior to incubation with appropriate secondary antibodies for 1 h at room temperature. The membranes were washed again and imaged using a Li-Cor Odyssey infrared imaging system.

### 2.5. Whole Blood Flow over a Collagen-Coated Surface

Whole blood was obtained from PDS25 and healthy control donors and anticoagulated using PPACK (Enzo) and Heparin (Sigma). Anticoagulated blood was perfused over a collagen-coated dish using a chamber from GlycoTech (Gaithersburg, MD, USA) under arterial (1000 s^−1^) shear rates. A Nikon Eclipse TE300 inverted microscope (200×) was used to observe thrombus formation and ImageJ (National Institutes of Health) was used for analysis.

### 2.6. Animal Housing and Production

Mice were housed in a pathogen-free facility, and P2Y12 D127N mice were produced by Cyagen Biosciences Inc. (Taicang, China) on a fee for service basis.

### 2.7. Tail Bleeding Assay

Mouse tail bleeding was conducted as previously described [23]. Mice aged 4–6 weeks were anesthetized prior to amputation of the distal 3 mm of the tail. The tail was then immersed in 37 °C saline and bleeding was monitored. If bleeding continued for greater than 600 s, then bleeding was halted manually by applying pressure.

### 2.8. Preparation of Mouse Platelets

Mouse blood was collected and platelets were isolated as previously described [22]. The resulting platelets were counted using a Hemavet 950FS blood cell analyzer (Drew Scientific, Dallas, TX, USA). Platelet counts were adjusted to a final concentration of 1.5 × 10^8^ cells/mL in N-2-hydroxyethylpiperazine-N′-2-ethanesulfonic acid-buffered (pH 7.4) Tyrode’s solution containing 0.2 U/mL apyrase.

### 2.9. Carotid Artery Injury

FeCl_3_ was used to injure the carotid artery as previously described [23]. Mice aged 10–12 weeks were anesthetized and the carotid artery was exposed. A baseline blood flow reading was obtained using a Transonic T402 flow meter (Ithaca, NY, USA). The carotid artery was injured using a 1 × 1 mm piece of filter paper saturated with 7.5% FeCl_3_ for 90 s. The filter paper was removed and blood flow was recorded.

### 2.10. Statistics

All statistical analysis was performed using Microsoft Excel and data was analyzed using a Student’s *t*-test where *p* < 0.05 was considered statistically significant. All data are presented as means ± STDEV of at least three independent experiments.

## 3. Results

### 3.1. ADP-Mediated Platelet Aggregation Is Disrupted in an Otherwise Healthy Human Subject

Platelet Defect Subject 25 (PDS25) is a 29-year-old male who originally enrolled as a healthy control donor. Blood cell counts from PDS25 were all within normal ranges, he reports no bleeding, and no spontaneous bruising. However, an evaluation of his platelets, which is common with new donors, revealed a strongly inhibited response to the agonist 2-MeSADP (Figure 1A). An analysis of Akt phosphorylation at serine 473, which occurs downstream of P2Y12 in a PI3K-dependent manner, showed a robust phosphorylation in healthy control platelets and a mild, though noticeable, phosphorylation in platelets from PDS25, which was eliminated with P2Y12 antagonism (Figure 1B). Analysis of PAR4-mediated platelet aggregation and secretion showed a defect in both aggregation and secretion at low doses of AYPGKF as one would anticipate if secondary signaling is inhibited (Figure 1C–E). Finally, there was no difference in aggregation and secretion when platelets from PDS25 were stimulated with a high dose of AYPGKF compared to similarly treated control platelets (Figure 1D,E). 

### 3.2. Defective VASP Phosphorylation and GTP-Rap1b Formation in Platelets from PDS25 

VASP is phosphorylated when cAMP is elevated via protein kinase A, and dephosphorylated when cAMP is squelched by G_i_-mediated signaling [24]. Given the observed defect in platelets isolated from PDS25, we evaluated VASP phosphorylation following stimulation with either 2-MeSADP or epinephrine in platelets from PDS25 and a control donor. Platelets were treated with PGE1 to raise cAMP production and thus raise phosphorylation of VASP. While subsequent treatment of control platelets with 2-MeSADP reduced phosphorylation of VASP, VASP phosphorylation in platelets from PDS25 remained elevated (Figure 2A,B). Contrarily, there was no difference in VASP phosphorylation between control and PDS25 platelets when epinephrine was used as an agonist suggesting that cAMP reduction downstream of G_z_ is intact. Consistently, we also observed inhibited GTP-Rap1b formation in platelets from PDS25 stimulated with 2-MeSADP compared to similarly treated control platelets (Figure 2C,D). The combination of 2-MeSADP and epinephrine enhanced the GTP-Rap1b signal, but it did not equal that of control platelets. These data suggest that cAMP and subsequent signaling downstream of P2Y12 is altered in platelets from PDS25 but is intact downstream of the epinephrine receptor.

### 3.3. Signaling Downstream of Gi-Coupled P2Y12 Is Inhibited in Platelets from PDS25

The data presented above suggest that platelets from PDS25 have a defect within the P2Y12 receptor or downstream signaling. Therefore, we isolated platelets from PDS25 and healthy control donors and determined whether aggregation via G_z_-coupled epinephrine was intact. Similar to G_i_, G_z_-mediated signaling results in suppressed cAMP production [25]. Epinephrine alone did not cause aggregation in platelets from either control or PDS25 (Figure 3A). As previously stated, stimulation with 2-MeSADP resulted in aggregation of control platelets, but only shape change of platelets from PDS25 (Figure 3A). However, when 2-MeSADP stimulation was supplemented with epinephrine control platelet aggregation increased while aggregation of platelets from PDS25 was equivalent to that of control platelets stimulated with 2-MeSADP alone (Figure 3A,B). These data suggest that signaling mediated by G_z_ is intact, while signaling mediated by G_i_ is perturbed in platelets from PDS25.

### 3.4. Thrombus Formation Is Reduced in Whole Blood from PDS25

To determine the consequence of the above observed responses of platelets from PDS25 to various stimuli on thrombus formation, we passed whole blood isolated from PDS25 and healthy control donors over a collagen coated surface at arterial shear rates. We found that thrombus formation was significantly inhibited in blood from PDS25 compared to healthy control blood (Figure 3C,D). These data suggest that thrombus formation is impaired in PDS25. The data presented above all point to a defect within the P2Y12 receptor. Therefore, we closely examined the P2Y12 receptor expressed by PDS25.

### 3.5. P2Y12 from PDS25 Contains a Novel SNP That Results in a D > N Substitution at Position 121

We sequenced P2Y12 from PDS25 to determine whether a mutation is present that may cause the observed phenotype. We amplified the coding sequence of P2Y12 in healthy donors and PDS25 using the following primers; sense 5′-TTAGAGGAGGCTGTGTCCAA-3′ and antisense 5′-GTCGTTTGTTTTGCTGCTAATA-3′ with an annealing temperature of 60 °C. We discovered a heterozygous G > A SNP at position 361 of the coding sequence of P2Y12 that predicts a D > N substitution at position 121 of the protein sequence of P2Y12. That aspartic acid residue is part of the DRY motif on P2Y12. The DRY motif is highly conserved among GPCRs and mutations within the DRY motif are known to influence GPCR function [26,27,28,29,30]. To study the effect of this SNP on in vivo platelet responses we produced a KI mouse with a similar D > N transition within the DRY motif of P2Y12. 

### 3.6. Reactivity to 2-MeSADP Is Altered in D127N P2Y12 Knock-In Mouse Platelets

Mouse P2Y12 contains additional amino acids so the DRY motif is at position 127 instead of 121 in human. Therefore, we produced D127N knock-in mice via CRISPR/CAS 9. The nucleotide substitution resulted in an AseI restriction enzyme cut site within the knock-in DNA. WT mice contain only the higher base pair product, which is cleaved by AseI resulting in two lower base pair products (Figure 4A). The heterozygous DNA contains all three products. P2Y12 D127N KI mice have no gross abnormalities and blood cell counts are comparable to WT mice (Table 1). Because we introduced a genetic alteration that should influence the DRY motif of P2Y12 we isolated platelets from WT, heterozygous D127N (Het), and homozygous D127 knock-in (KI) mice and analyzed aggregation in response to the P2Y12 (and P2Y1) agonist 2-MeSADP. We found that aggregation in response to 10 nM 2-MeSADP was significantly reduced in platelets from Het mice, but completely abolished in platelets from homozygous D127N KI mice (Figure 4B,C). When platelets were stimulated with 100 nM 2-MeSADP aggregation was not different between WT control and Het mice, but aggregation of D127N KI platelets was significantly reduced compared to WT (Figure 4D,E). Additionally, we stimulated washed platelets from WT, Het, and KI mice with the physiological agonist ADP. We found that aggregation was greatly reduced in KI platelets compared to WT platelets when either 5 μM or 10 μM ADP was used as an agonist (Figure 5A–C). Further, we stimulated washed platelets isolated from WT, Het, or KI mice with two concentrations of AYPGKF and measured aggregation and dense granule secretion. We found that aggregation was unaltered in Het or KI mouse platelets compared to WT, but dense granule secretion was reduced in KI platelets when a low concentration of AYPGKF was used (Figure 5D–I). These data are consistent with impaired reactivity to ADP in the Ki mice.

### 3.7. Hemostasis and Thrombosis Are both Interrupted in P2Y12 D127N Knock-In Mice

To determine whether hemostasis is altered by the D127N transition we performed tail bleeding experiments in WT, Het, and homozygous KI mice. We found that there was no difference in tail bleeding times between WT and Het (Figure 6A). However, tail bleeding had to be stopped mechanically in all KI mice. Therefore, an intact DRY motif within P2Y12 is crucial for the maintenance of hemostasis. To determine the effect of a disrupted DRY motif in P2Y12 on thrombosis we injured the carotid artery of WT, Het, and homozygous KI mice with 7.5% FeCl_3_ for 90 s and monitored blood flow. Time to occlusion was significantly enhanced in Het mice compared to WT control mice, suggesting that even a partial loss of WT P2Y12 can have an impact on thrombus formation (Figure 6B,C,E). Consistently, the carotid artery in all but 2 KI mice failed to occlude over the 30 min experiment (Figure 6D,E). Additionally, we noted unstable thrombus formation in 88.9% of KI mice, 81.9% of Het mice, and 11.1% of WT control mice. These data agree with our bleeding time data and suggest that the DRY motif on P2Y12 is essential for its function. 

## 4. Discussion

In this report we reveal a novel SNP within human P2Y12 that results in a D > N substitution in the DRY motif. We report that the subject (PDS25), who is heterozygous for this substitution, has no history of bleeding, but his platelets do not aggregate in response to 2-MeSADP.

To our knowledge the mutation described in this report is novel. However, there are numerous reports detailing variants within the P2Y12R gene [16,17,31,32,33]. Many of these result in deletion or reduced expression of P2Y12. In fact, our observation that platelets from PDS25 do not aggregate in response to 2-MeSADP is in agreement with a previous report [17]. In that report the authors describe an R122C substitution, which is also within the DRY motif. In their case however, the subject was homozygous for the mutation and had associated bleeding. It should be noted that the R122C subject also had an associated rare SNP within PAR1 that results in lower PAR1 expression, though this would not impact ADP-mediated aggregation. PDS25 is heterozygous for the mutation and has no history of bleeding. Further, platelets from the R122C subject did not aggregate in response to ADP, which is similar to the phenotype observed when platelets from PDS25 are stimulated with 2-MeSADP. It is interesting that platelets from two family members of the R122C subject were also tested and both had at least mild aggregatory responses to low dose ADP. Both family members were heterozygous. This suggests that the aspartic acid and the arginine within the DRY have separate functions, as PDS25 had no aggregatory response to high dose 2-MeSADP.

The primary reason for generating a knock-in mouse model of D127N P2Y12 is to determine whether the mutation affects thrombosis or hemostasis or both. It is well known that P2Y12 antagonism protects patients from thrombosis but may result in bleeding. Therefore, we expected to see a loss of hemostasis and interrupted thrombosis in homozygous D127N knock-in mice, which is what we observed. We were more interested in the phenotype of the heterozygous mice because our subject is heterozygous for the mutation. Furthermore, our subject has not experienced excess bleeding. Indeed, tail bleeding, a measure of hemostasis, was not altered in heterozygous D127N mice. However, occlusion following FeCl_3_-induced injury was prolonged and thrombus stability was reduced in heterozygous D127N mice compared to WT control mice. This is notable because bleeding is often associated with P2Y12 antagonism and because the isolated platelet phenotype resulting from the D121N mutation in our subject is severe. That said, we were surprised by the reactivity of platelets isolated from heterozygous D127N mice. 

Isolated platelets from the heterozygous D127N mouse do not phenocopy platelets isolated from PDS25. While we were unable to measure any appreciable aggregation of platelets from PDS25 when stimulated with 100 nM 2-MeSADP, we were able to measure aggregation in heterozygous knock-in mice at 10 nM 2-MeSADP and aggregation resulting from 100 nM 2-MeSADP was not different from WT control. We believe the explanation for this discrepancy may be linked to the copy number of P2Y12 on mouse and human platelets. A report in 2013 suggests that humans have 425 ± 50 copies of P2Y12 on the platelet surface, while mouse platelets contain 634 ± 87 copies of P2Y12 [34]. The approximate 50% increase in P2Y12 copy number in mouse is magnified by the small size of the mouse platelet. In PDS25 we were able to observe an increase in pAkt S473 following 2-MeSADP stimulation, and this was abolished with AR-C69931MX pretreament suggesting that there is some signaling originating from P2Y12 in platelets from PDS25. It is possible that there are not enough WT copies of P2Y12 available to induce aggregation of platelets from PDS25, but there are enough WT copies over a smaller surface area of platelets from D127N heterozygous platelets. It is not yet clear whether oligomerization, heterodimerization, or G-protein coupling is the reason for the lack of aggregation in PDS25, but that would be an interesting area of exploration.

Others have mutated the DRY motif in other GPCRs. In most cases the mutated GPCR becomes constitutively active [28,29,35,36,37,38]. There are, however, those that develop constitutive inactivity [27,39,40]. We appear to have the latter phenotype with PDS25, but that is not necessarily the case. The aspartic acid in the DRY motif is thought to regulate receptor conformation, the mutation of which may allow the receptor to adopt an active conformation. Interestingly, this active conformation does not always equate to constitutive activity. For instance, in the TP receptor an E129V (DRY can also be ERY) mutation resulted in greatly enhanced agonist binding, but the mutation is classified as constitutively inactive [27]. The same can be said of other similar mutations in some other GPCRs [39,41,42]. Therefore, at this time, it is unclear which category D121N would fit.

Here, we show a novel P2Y12 variant that results in a D < N transition within the DRY motif. Given that PDS25 has never had a bleeding complication of any sort, and that inhibition of P2Y12-mediated platelet aggregation is greater than one would see in platelets isolated from a patient treated with a commercial P2Y12 antagonist, we believe there is a tremendous opportunity here for discovery.

## Figures and Tables

**Figure 1 ijms-23-11519-f001:**
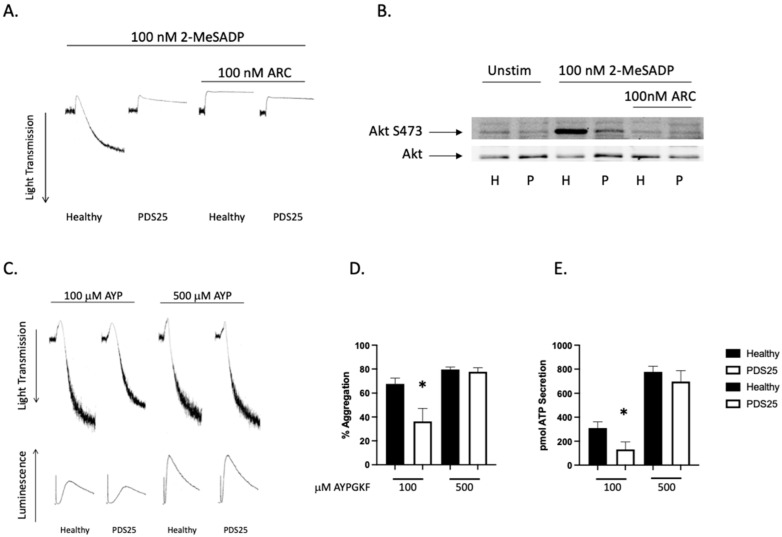
2-MeSADP-mediated aggregation is interrupted in platelets from PDS25. (**A**) Representative tracings showing aggregation of platelets from PDS25 and healthy control donors that were stimulated with 100 nM 2-MeSADP following pretreatment of 100 nM AR-C69931MX or not and monitored for 3 min to observe aggregation. (**B**) Western blot showing Akt phosphorylation in platelets from PDS25 or a healthy control following stimulation with 2-MeSADP. Total Akt was used as a loading control. (**C**) Representative tracings showing aggregation and secretion of platelets from a healthy control donor and PDS25 stimulated with the indicated concentration of AYPGKF. (**D**) Quantitation of aggregation of platelets from healthy control donors and PDS25 stimulated with the indicated concentration of AYPGKF. (**E**) Quantitation of ATP secretion from platelets stimulated with the indicated concentration of AYPGKF. * *p* < 0.05 vs. healthy control, *n* = 5.

**Figure 2 ijms-23-11519-f002:**
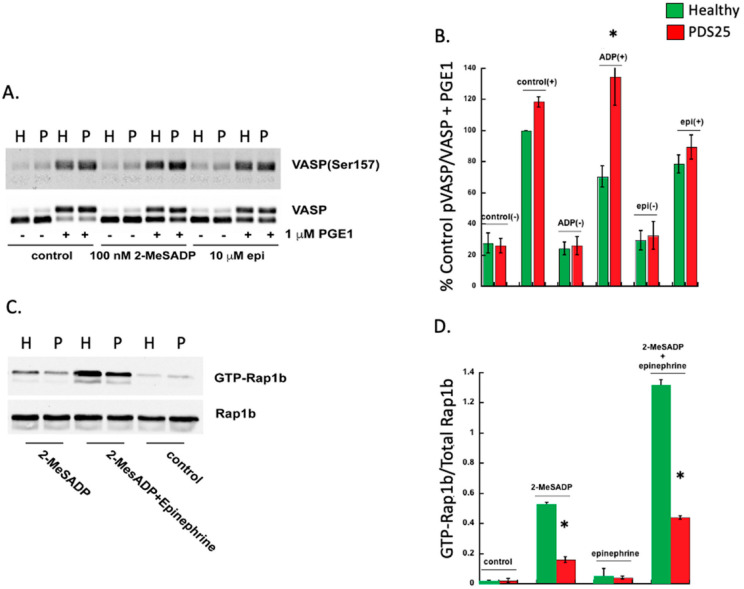
VASP phosphorylation is enhanced, while Rap1b-GTP formation is reduced in platelets from PDS25 stimulated with 2-MeSADP. (**A**) Representative Western blot showing VASP phosphorylation in response to either 100 nM 2-MeSADP or 10 mM epinephrine for 3 min with or without PGE1 pretreatment. Total VASP was used to assess loading. (**B**) Quantitation of VASP phosphorylation as described in A. (**C**) Representative Western blot showing Rap1b-GTP in platelets from PDS25 or normal control donors stimulated with 100 nM 2-MeSADP alone or in combination with 10 μM epinephrine. Rap1b was used to assess loading. (**D**) Quantitation of GTP-Rap1b expressed as a ratio of total Rap1b as seen in C. H = Healthy control, P = PDS25. * *p* < 0.05 vs. control, *n* = 3.

**Figure 3 ijms-23-11519-f003:**
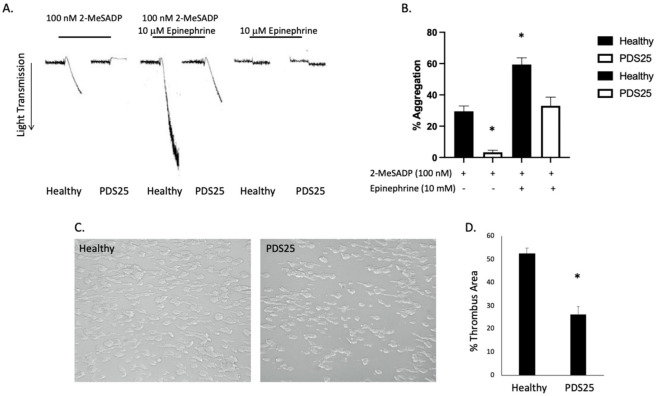
G_z_-mediated aggregation is intact in platelets from PDS25, while thrombus formation on collagen is inhibited in whole blood from PDS25. (**A**) Representative aggregation tracings of platelets isolated from PDS25 or healthy controls stimulated with 100 nM 2-MeSADP, 10 μM epinephrine, or both. (**B**) Quantitation of aggregation as represented in panel A. * *p* < 0.05 vs. healthy control stimulated with 100 nM 2-MeSADP, *n* = 4. (**C**) Representative images of thrombus formation following whole blood (from PDS25 or healthy controls) flow over a collagen-coated surface. (**D**) Quantitation of thrombus area resulting from whole blood flow over collagen. * *p* < 0.05 vs. healthy control, *n* = 5.

**Figure 4 ijms-23-11519-f004:**
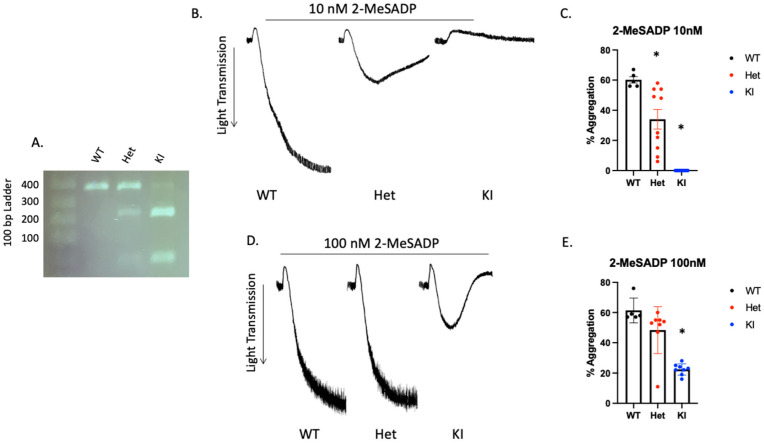
2-MeSADP-mediated platelet aggregation is interrupted in platelets from P2Y12 D127N knock-in mice. (**A**) Representative agarose gel showing the banding pattern of WT, heterozygous (Het), and homozygous D127N knock-in (KI) mice. (**B**) Representative tracings showing WT, Het, and KI platelet aggregation in response to 10 nM 2-MeSADP. (**C**) Quantitation of platelet aggregation in response to 10 nM 2-MeSADP as described in (**B**). (**D**) Representative tracings showing WT, Het, and KI D127N platelet aggregation in response to 100 nM 2-MeSADP. Quantitation of platelet aggregation in response to 10 nM 2-MeSADP as described in (**E**). * *p* < 0.05 vs. WT, *n* = 5.

**Figure 5 ijms-23-11519-f005:**
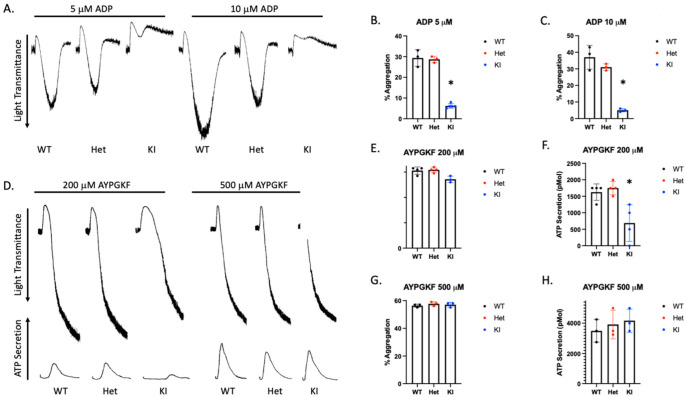
D127N KI mouse platelets have a significantly inhibited response to ADP. (**A**) Representative tracings showing aggregation in response to either 5 or 10 μM ADP of platelets isolated from WT, Het, or KI mice. (**B**,**C**) Quantitiation of platelet aggregation in response to 5 or 10 μM ADP. (**D**) Representative aggregation and secretion tracings of platelets isolated from WT, Het, and KI mice stimulated with 200 or 500 μM AYPGKF. (**E**,**F**) Quantitation of aggregation (**E**) and secretion (**F**) from WT, Het, and KI platelets stimulated with 200 μM AYPGKF. (**G**,**H**) Quantitation of aggregation (**G**) and secretion (**H**) from WT, Het, and KI platelets stimulated with 500 μM AYPGKF.

**Figure 6 ijms-23-11519-f006:**
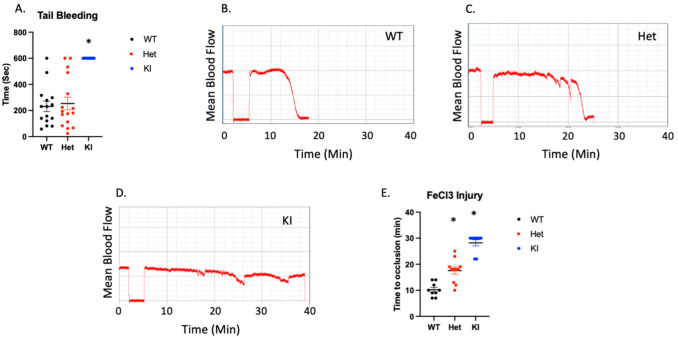
Hemostasis and thrombosis are interrupted in P2Y12 D127N knock-in mice. (**A**) Time for bleeding to stop following excision of the distal 3 mm of the tail. (**B**–**D**) Representative images showing blood flow in the carotid arteries of WT, Het, and KI mice following administration of 7.5% FeCl_3_. (**E**) Quantitation of time to vessel occlusion following FeCl_3_ injury to the carotid artery. * *p* < 0.05 vs. WT using Wilcoxon test, *n* = 9–11.

**Table 1 ijms-23-11519-t001:** Blood cell counts in WT, Heterozygous D127N, and Homozygous D127N knock-in mice.

Parameter	WT	Heterozygous D127N	Homozygous D127N
WBC (K/mL)	5.67 ± 1.25	6.04 ± 0.80	5.06 ± 0.80
NE (K/mL)	0.83 ± 0.23	0.59 ± 0.08	0.69 ± 0.09
LY (K/mL)	4.50 ± 1.00	4.86 ± 0.66	3.61 ± 0.70
MO (K/mL)	0.44 ± 0.11	0.59 ± 0.10	0.54 ± 0.12
RBC (10^6^/mL)	9.05 ± 0.20	9.29 ± 0.25	9.30 ± 0.48
Hct (%)	38.76 ± 0.94	40.60 ± 1.14	40.28 ± 1.81
Plt (K/mL)	761 ± 29	721 ± 33	720 ± 56
MPV (fL)	4.17 ± 0.06	4.12 ± 0.02	4.22 ± 0.08

## Data Availability

All data is available within the article.

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
