# Peer review of "D121 Located within the DRY Motif of P2Y12 Is Essential for P2Y12-Mediated Platelet Function"

_ijms, 2022, doi:10.3390/ijms231911519_

Round 1
Reviewer 1 Report
In the present manuscript the author assessed platelet function from one male patient (29-year-old) who presented a heterozygous mutation of D121N within the DRY motif. According to the authors, the mutation described in this manuscript is novel. Briefly, the aggregation magnitude induced by the P2Y12 agonist, 2-MeSADP (100 nM) and the PAR agonist, AYPGKF (100 uM) were significantly reduced in platelets from this patient in comparison with the control. Similar finding was observed in the downstream signalling pathway of P2Y12 activation where Akt phosphorylation at Ser243 was reduced in platelets from this patient. On the other hand, the phosphorylation of VASP at Ser157, a marker of cAMP accumulation was significantly greater while Rap1b-GTP formation is reduced in platelets stimulated with 2-MeSADP. A heterozygous D127N (Het) and homozygous D127 knock-in (KI) mice that mimic this mutation was generated and similar findings in platelet reactivity was observed, that is a reduction in response to ADP (5 ou 10 uM) and to the 2-MeSADP agonist. Tail bleeding had to be stopped mechanically in knock-in mice. The time to occlusion following FeCl3 injury was significantly greater in Het (20 min) and KI (30 min) in comparison with wild-type (10 min) mice. Please find below some minor suggestions:
Introduction
On page 2, lines 46-48: please insert references for this statement
Results
Figure 2A: Is there a better representative image as the intensity of the band for p-VASPSer157 in platelets from patient stimulated with 2-MeSADP seems not to represent the results presented on figure 2B.
Reviewer 2 Report
Congratulation for your work! I consider the article valuable and scientifically sound.
I would have only one minor observation: how many mice were used? I would predict a small number - therefore results should be reported as median, not as mean.
Author Response
We appreciate the reviewer's kind message. Thank you for your encouragement!
Question: how many mice were used? I would predict a small number - therefore results should be reported as median, not as mean.
Response: Mouse numbers are reported in each figure legend except where each value is visible in a bar graph. We have routinely reported mouse data as they are reported in the manuscript. However, we would be happy to defer to the journal statisticians for guidance in this manner. If the editors or statisticians would like us to alter our data as to present median values, then we will do that.